# Estimates of Genomic Heritability and the Marker-Derived Gene for Re(Production) Traits in Xinggao Sheep

**DOI:** 10.3390/genes14030579

**Published:** 2023-02-25

**Authors:** Zaixia Liu, Shaoyin Fu, Xiaolong He, Xuewen Liu, Caixia Shi, Lingli Dai, Biao Wang, Yuan Chai, Yongbin Liu, Wenguang Zhang

**Affiliations:** 1College of Animal Science, Inner Mongolia Agricultural University, Hohhot 010018, China; 2Inner Mongolia Engineering Research Center of Genomic Big Data for Agriculture, Hohhot 010018, China; 3Institute of Animal Husbandry, Inner Mongolia Academy of Agricultural and Animal Husbandry Sciences, Hohhot 010031, China; 4College of Agronomy, Animal Husbandry and Bioengineering, Xing’an Vocational and Technical College, Ulanhot 137400, China; 5Veterinary Research Institute, Inner Mongolia Academy of Agricultural and Animal Husbandry Sciences, Hohhot 010031, China; 6School of Life Science, Inner Mongolia University, Hohhot 010021, China

**Keywords:** Xinggao sheep, LS, BWT, WWT, ADG, genetic parameter evaluation, MATs

## Abstract

Xinggao sheep are a breed of Chinese domestic sheep that are adapted to the extremely cold climatic features of the Hinggan League in China. The economically vital reproductive trait of ewes (litter size, LS) and productive traits of lambs (birth weight, BWT; weaning weight, WWT; and average daily gain, ADG) are expressed in females and later in life after most of the selection decisions have been made. This study estimated the genetic parameters for four traits to explore the genetic mechanisms underlying the variation, and we performed genome-wide association study (GWAS) tests on a small sample size to identify novel marker trait associations (MTAs) associated with prolificacy and growth. We detected two suggestive significant single-nucleotide polymorphisms (SNPs) associated with LS and eight significant SNPs for BWT, WWT, and ADG. These candidate loci and genes also provide valuable information for further fine-mapping of QTLs and improvement of reproductive and productive traits in sheep.

## 1. Introduction

Sheep domestication began approximately 9000 years ago in southwestern Asia, in today’s Iran and Turkey [1]. China has a large animal husbandry scale and an abundance of ovine germplasm, accounting for approximately 10% of the world’s 700 breeds [2], 42 indigenous sheep breeds, 30 cultivated breeds and supporting lines, and 8 introduced breeds and supporting lines [3,4]. China’s local breeds have the advantages of crude feed tolerance, strong adaptability, and low maintenance requirements, but the growth rate is slow, the reproductive performance is poor, the performance measurement is lagging, breeding intensity is not adequate, and population genetic progress is slow. The Xinggao sheep is a Chinese domestic sheep breed made by crossing Hu sheep ewes as the female parent and Friesian sheep as the male parent through crossbreeding and transverse cross fixation, which has been promoted on a large scale in the Hinggan League and has promoted the transformation and upgrading of the local mutton sheep industry [5]. Hu sheep are renowned for their high adaptability, high reproductive performance, long period of estrus, the advantages of having multiple lambs per fetus, rapid growth, development, and are widely reared on the Taihu Plain of China [6,7]. East Friesian sheep are a breed of sheep known worldwide for both milk and meat, native to Germany and Holland [8]. The average lambing rate of the ewe is up to 200–230%, and milk yield can reach 500–810 kg [9]. East Friesian sheep have a good performance in lactation and reproductive and growth performance, but the disadvantage is poor adaption. For this reason, we prefer to crossbreed Hu sheep with East Friesian sheep to increase diversity and breed high-quality meat suitable for the highly cold climate characteristics of the Hinggan League. Crossbreeding programs for traits such as growth, development, and meat quality in sheep are developing rapidly.

Improved reproductive and growth performance of ewes and lambs have been associated with greater profitability and life cycle efficiency. Individual birth weight, body growth, and reproductive traits are vital indicators in litter-bearing animals and increase productive efficiency [10,11]. Genetic assessment is used to select traits for populations, predict the direct and correlated responses to selection, screen a breeding system for future improvement, and estimate genetic gain. A variety of genetic parameters have been estimated in animals [12,13,14,15]. However, the estimation of genetic parameters for traits is not fixed; it may vary from population to population and may change with environment and time, etc. The estimation of genetic parameters in Xinggao sheep and the identification of the best models for this population phenotype are crucial for improving the genetic benefits of economically important traits. Reproductive traits such as ovulation rate, litter size, age at first lambing, stillbirth, lamb weight and weaning, and daily weight gain have all been shown to have a significant impact on the economics of sheep farming [16]. Among them, increasing litter size can improve sheep production efficiency. Ewes (female sheep) generally give birth to a single lamb, with the phenomenon being controlled by both genetic and environmental factors [17]. Prolificacy or reproductive traits are typically regulated by different genes in sheep with different effects [18]. The *FecB* gene was the first to be identified as a major gene that has been shown to affect fertility [19]; this was followed by the discovery and verification of *BMP15* [20], *GDF9* [21], and *B4GALNT2* [22] genes. GWAS identified that several genes associated with the size and functions of the litter have been reported in different breeds of sheep [23]. There is weak positive selection acting on reproduction and growth in a sheep population, with Xinggao sheep having higher fitness and prolificacy on average, across which we wish to estimate heritability. We genotyped Xinggao sheep using GGP Ovine 50K and attempted to describe the genetic heritage given to sheep genomes by selection and adaptation.

## 2. Materials and Methods

### 2.1. Animals Used and Obtaining Phenotypes

Our study utilized Xinggao sheep, with a record of all three consecutive births (from 2018 to 2020) at the Inner Mongolia DuMei pasture Bio Technology Co., Ltd. Jalaid Banner, Hinggan League, China (46°04′–47°21′ N, 121°17′–123°38′ E). Xinggao sheep are characterized by a high reproduction rate, rapid growth and development, high resistance to cold, suitability for captive housing, and good nutritional benefits and are a new breed of mutton sheep suitable for confined feeding on a largescale in northeastern regions of Inner Mongolia. In 2018, 618 ewes were mated to 5 sires to produce 1186 lambs; in 2019, 380 ewes mated to 6 sires gave birth to 867 lambs; and 404 ewes mated with 5 sires in 2020 to produce 930 lambs. Male and female lambs are produced annually at a ratio of 1:1. To remove outliers during data statistics, the following methods were used: (1) the first and third quartiles for each trait were calculated, (2) the interquartile range was assessed, (3) the upper and lower bounds of the data range were returned, and (4) data below the lower limit or above the upper limit were outliers. After the removal of outliers and missing data, the pedigree contained a total of2289lambs born to 11 sires and 946 dams. Data on the reproductive traits of ewe on litter size (LS) came from the spring (3–5 months, (1)), summer (6–9 months, (2)), and autumn (10–11 months, (3)) of these years. Lamb’s three growth performances were recorded: birth weight (BWT), weaning weight (WWT), and average daily gain (ADG). Among them, WWT was used to linearly adjust for weaning age at 50 days and to record lamb wean days (WD). The weaning weight formula corrected to 50 days was: $WWT=\frac{WWT-BWT}{WD}\times 50d+BWT$.

Phenotypes records for Xinggao ewes and lambs were gathered, and there were 2289 records of the reproductive trait litter size (LS) in 946 ewes and three growth traits for lambs: birth weight (BWT), weaning weight (WWT), and average daily gain (ADG), which had 2281, 2284 and 2284 phenotypic records, respectively (Table 1), which were used in the final analysis of the fitted mixed models. With our expectations, the mean of LS records was 2.12 and ranged from 1 to 5, with Xinggao sheep being lambs of multiple births.

### 2.2. Statistical Analysis 

Genetic parameters are principal components in any breeding program for selecting animals based on their phenotypic performance. The multivariate mixed models (DMU) program was used to evaluate the genetic parameters of re(production) and estimated breeding values (EBV)using the restricted maximum-likelihood algorithm [24,25]. To determine the variance, (co)variance components, and correlations for the attributes of the ewe and lamb, respectively, single-trait and multi-trait animal models were utilized. Permanent environment, animal genetics, residuals, and mating sire were analyzed as a random effect influences on the reproductive traits. Lambing season (1, 2, 3), and the year of calving (2018, 2019, 2020) were fixed effects for LS, and the module was [26]:Y=Xb+Za+Wpe+Ts+e
where *Y* is the LS observation; *b* is fixed effects; *a* is the vector of solutions for the coefficients of direct animal (additive) genetic random effects; *pe* is the vector of solution for permanent environmental effects; *s* is the random effect of the service sire in the year of calving; *e* is the vector of residual effects; *X*, *Z*, *W*, and *T* are the correspondent incidence matrices of the fixed effects and additive genetic, permanent environmental random effects due to repeated records per ewe and random effect of the service sire, respectively.

Three traits were analyzed for lamb performance: BWT, WWT, and ADG, with the permanent environment, animal genetics, maternal genetics, and residuals as random effects. Fixed effects included the following variables: lambing year (three levels: 2018, 2019, 2020); lambing season (three levels: 1, 2, 3); and gender (two levels: male and female). The module was:Q=Eb+Fa+Hpe+Km+e
where, *Q* is the observed BWT, WWT and ADG, *b* is the fixed effects; *a* is the random direct effects; *pe* is permanent environmental random effects; *m* is the maternal genetic effects; *e* is the random residual. *E*, *F*, *H* and *K* are incidence matrices of the fixed effects, additive genetic random effects, permanent environmental effects, and maternal genetic random effects, respectively.

### 2.3. Estimation of Genetic Parameters

Heritability of the LS trait was estimated as the ratio of the additive genetic variance to total phenotypic variance; repeatability as the ratio of the sum of the additive genetic variance and permanent environmental variance to phenotypic variance [24].
$$h^{2}=\frac{\sigma_{a}^{2}}{\sigma_{a}^{2}+\sigma_{pe}^{2}+\sigma_{s}^{2}+\sigma_{e}^{2}}\;\;\;r=\frac{\sigma_{a}^{2}+\sigma_{pe}^{2}}{\sigma_{a}^{2}+\sigma_{pe}^{2}+\sigma_{e}^{2}}$$
where *h*^2^ is heritability; *r* is repeatability; *σ*^2^*_a_* is direct additive genetic variance; *σ*^2^*_pe_* is permanent environment variance related to repeats; *σ*^2^*_s_* is service sire in the year of calving random variance; and *σ*^2^*_e_* is residual variance.

BWT, WWT, and ADG were used to estimate the variance component for heritability and genetic correlation.
$$h^{2}=\frac{\sigma_{a}^{2}}{\sigma_{a}^{2}+\sigma_{pe}^{2}+\sigma_{m}^{2}+\sigma_{e}^{2}}$$
where *h*^2^ is heritability; *σ*^2^*_a_* is direct additive genetic variance; *σ*^2^*_pe_* is permanent environment variance related to repeats; *σ*^2^*_m_* is maternal genetic variance; and *σ*^2^*_e_* is residual variance. EBV for 4 traits and the top 500 breeding values were drawn in a Venn diagram for 4 traits [27].

### 2.4. Genotypic Data

We first chose 49 ewes and 46 lambs for genotype testing in order to perform an in-depth study on Xinggao sheep. At LS 1 to 4, 49 ewes were selected for blood collection. Blood tests taken from 46 healthy and active lambs revealed that the ranges for birth weight, weaning weight, and average daily growth were 2.3 kg to 6.3 kg, 10.2 kg to 21.51 kg, and 0.14 kg to 0.37 kg, respectively. DNA was extracted from 200μL of blood using the Quick-DNA-Miniprep kit (Zymo Research, Irvine, CA, USA) according to the manufacturer’s protocol. The DNA obtained was purified using the DNA Clean & Concentrator-5 kit (Zymo Research). The DNA concentration was measured with a Nanodrop Lite (Thermo Scientific^®^, Wilmington, DE, USA), and integrity was visualized on a 1.5%agarose gel. The 95 sheep were genotyped using a medium-density GGP Ovine 50K array (51,867 SNPs) in the NEOGEN Genomics. In the future, we will follow up on this batch of sheep and perform more comprehensive related tests to verify.

### 2.5. Genotyping Analysis and Data Quality Control

A total of 51,867 SNPs were obtained from genotyping data. Quality control (QC) of genotyping data was performed using the PLINK v1.9 software [28] to filter out those SNPs and individuals that failed using the following specific filtering criteria: 51,867 SNPs and individuals with call rate < 95% (−geno:0.05 and −mind:0.05),minor allelic frequency (MAF) < 0.01, and which failed the Hardy–Weinberg equilibrium (HWE) (*p* < 1 × 10^−6^) were excluded [29]. Finally, genotype association tests were performed on 45,147 SNPs distributed on 27 chromosomes in 49 ewes and 46 lambs.

### 2.6. Genome-Wide Association Study

The TASSEL 5.2.81 software was used to analyze LS, BWT, WWT, and ADG traits as well as high-quality SNPs for genome-wide association study (GWAS) [30] and to identify the marker-trait associations (MTAs) employing linear model (GLM). In general, the GLM models focus on the SNP effects, which contain fixed effects such as population structure and genotypes. The quantile–quantile (Q-Q) plots were used to select the best model for each trait. Bonferroni correction was applied to control for error rate, and chromosome-wide significance thresholds of 5% (0.05/(45147/27 = 2.99 × 10^−5^)) were used for calculation, the chromosome-wide significance level was used as the criterion to call significant association, and a suggestive association corresponds to a *p* < 10^−4^ [31]. Association maps and the significant SNPs are visualized with a threshold line in a Manhattan plot. The Manhattan plot and Q-Q plot were plotted using R v. 4.1.2 [32]. The calculation formula of inflation factor (λ) [33] is: (a) *p*_value = gwas$*p*, (b) z = qnorm(*p*_value/2), (c) λ= round(median(z^2^, na.rm = TRUE)/0.454, 3).

### 2.7. Identifying Candidate Gene

Potential candidate genes were located within 50kb upstream and downstream from the detected significant SNPs using Oar_v4.0 sheep reference genome from the NCBI’s Genome (https://www.ncbi.nlm.nih.gov/genome/gdv/?org=ovis-aries, accessed on 20 October 2022).

## 3. Results

### 3.1. Genetic Parameter Estimates for Phenotypes

There were approximately equal numbers for these four traits, and all recorded lambs had complete production traits (Table 1). In terms of the 946 ewes, they gave birth to an average of 2.12 lambs; most lamb weights at birth were between 2.5 kg and 4.0 kg, and the weaning weight was between 12 kg and 15 kg.

The heritability LS was 0.12, and the repeatability was 0.15. The heritability of BWT, WWT, and ADG were 0.22, 0.28, and 0.29, respectively; the genetic correlation between BWT and WWT was 0.34, and the genetic correlation between WWT and ADG was the highest at 0.97, and for BWT and ADG, the value was 0.12 (Table 2). The high genetic correlation between WWT and ADG is high since most variants in ADG are associated with WWT rather than BWT. WWT had high phenotypic correlation with ADG, while BWT had the lowest phenotypic correlation with ADG at 0.13, which accounts for the fact that these traits are more or less influenced by the same genetic and environmental factors.

Breeding values of ewe reproduction and lamb production performance were estimated with year and sex as fixed effects. Over the years, the EBVs (estimated breeding values) of LS, BWT, and ADG have been basically floated within 3 years, and WWT has gradually increased, with EBV reaching18.23 ± 0.28 in 2020. Male lambs were found to have a higher EBV than female lambs (Figure 1).

The top 500 EBVs for each trait were sorted to select the most favorable lambs for breeding (Figure 2), including 26 males and 32 females, ranked for the 3traits. Average BWT, WWT, and ADG values were 4.46 kg, 20.54 kg, and 0.32 kg, respectively, which is higher than the average level for the whole flock. 

### 3.2. Genome-Wide Association Study

GWAS analysis of re(production) traits in ewes and lambs was performed. A total of 45,147 SNPs on 27 sheep autosomes were selected for association analyses. These SNPs passed the quality control criteria. The genome-wide scan revealed a significant association between SNPs and BWT, WWT, and ADG (Table 3, Figure 3). For the LS, the associations did not cross the significance threshold, though there was suggestive evidence (Table 3, Figure 3A), and the 2 SNPs (oar3_OAR14_38652542 and OAR25_39236034.1) were suggestively significant (*p* < 10^−4^). The Q-Q plot showed the total distribution of the observed *p* (−log10 *p*) of 45,147 SNPs versus the expected values, showing that some deviated from the expected with an inflation factor (λ) of 1.002. Of the lambs’3growth traits, a total of 8 SNPs on 5 chromosomes crossed the chromosome-wide significance thresholds (*p* < 2.99 × 10^−5^), and 24 were suggestively significant SNPs (*p* < 10^−4^) (Figure 3). Out of these, 2, 3, and 3 SNPs showed genome-wide significance associated with BWT (Figure 3B), WWT (Figure 3C), and ADG (Figure 3D), respectively, and the Q-Q plot inflation factor (λ) was greater than 1.

## 4. Discussion

### 4.1. Genetic Parameter Estimates for Phenotypes

Reproduction is a key factor in breeding sheep and herd expansion. Fecundity is a complicated trait reflecting differences in rates of ovulation/fertilization, the uterine environment, and embryo/fetus survival. Heritability mainly reflects the magnitude of the additive effects of gene and is a comprehensive reflection of traits, populations, and environments [34]. Therefore, the estimated heritability of traits generally has certain deviations, but there is still relative stability. It is necessary to improve the accuracy of heritability estimation in breeding planning. Reproductive traits typically have low to moderate heritability; that is, phenotypic selection was limited and gives rise to a slow genetic gain [35]. In univariate animal models of repeatability, the reproductive trait LS of the Xinggao ewe was analyzed in the current study’s analysis. When the direct and maternal genetic effects as well as the service sire in our study were taken into account, the heritability and repeatability for LS were 0.12 was 0.15, respectively. We found maternal heritability and maternal permanent environmental effects to be significant for LS. This result differs from studies in Romanov sheep reported by Murphy et al., where phenotypic variance (σ^2^_P_) in litter size was low due to additive inheritance (0.06 to 0.08) and the influence of the permanent environment (0.05 to 0.07) in ewes [36]. Masood reported that in a linear model, the EBV of line S (beginning and end of the joining period for selection) fertility increased from −0.022 ± 0.040 in 1989 to 0.153 ± 0.044 in 2005; following a period of an out-of-season spring joining, the regression coefficient for EBV was 0.0101 ± 0.0004/year and the estimated heritability of the litter size was 0.04 ± 0.02, and the repeatability was 0.12 ± 0.03 [26]. For the heritability (0.08 to 0.09) and repeatability (0.10 to 0.12) for the number of lambs born on a per ewe lambing (NLBL) in Czech Romanov [37,38] and in Australian Merino sheep, the heritability of the litter size was 0.074 [39]. In Suffolk and Texel sheep, litters born after hormonal-induced estrus and after natural estrus were estimated to have a heritability of between 0.05 and 0.18 [40]. In Egyptian goats, using three animal models, estimate direct (additive) and maternal genetic effects for litter size at birth), heritability was 0.06 to 0.19 [41]. For Targhee sheep, using single-trait analyses, heritability estimates were 0.10 for litter size at birth and 0.07 for litter size at weaning [42]. Through the above comparative analysis, we can find that the heritability of LS is different due to different treatments, different models, and different breeds. In general, reproductive traits tend to be traits with low heritability. 

We selected three growth traits for lamb performance, namely BWT, WWT, and ADG, which were analyzed in animal models with multi-traits considering the permanent environment, animal genetics, and maternal genetics. In this study, the heritabilities of BWT, WWT, and ADG were 0.22, 0.28, and 0.29, respectively. There is a certain relationship between the traits shown by organisms. When a trait is selected, it also indirectly produces selection effects on other traits. Generally, genetic correlation and phenotypic correlation are used to measure the degree of correlation traits. The genetic correlation of BWT was 0.34 with WWT and 0.12 with ADG. The genetic correlation was 0.97 between WWT and ADG, indicating that WWT may be able to effectively increase ADG. The phenotypic correlation between the WWT and ADG traits was 0.97, and the lowest correlation was 0.13 between BWT and ADG. The results indicated that there were different degrees of correlation between the growth traits of Xinggao sheep. Others have reported in sheep heritabilities of the birth weight and body weight of lambs at 60, 90, and 120 days of age ranged from 0.015 to 0.19 [26]. Hanford et al. [42] reported direct heritabilities of 0.25 for birth weight and 0.22 for weaning weight as well as maternal heritability of 0.20 for birth weight and 0.11 for weaning weight, and between the birth and weaning weights of direct genetic correlation it was 0.52. Khattab et al. [43] reported heritabilities of 0.14 for birth weight by the single-trait animal model and 0.17 by the multi-trait animal model in Rahmani and Romanov Sheep. Lalit et al. [44] reported heritabilities of 0.38 for birth weight and 0.45 for weaning weight in Harnali sheep. Analysis of estimated breeding values (EBVs) for BWT, WWT, and ADG showed that the sex had differential effects, with males being heavier than females for all three traits. The direct autosomal heritabilities for birth weight and weaning weight were 0.12 and 0.29, respectively, and the sex effect was significant (*p* < 0.01) across all traits [45], and a similar result was observed. The effect of lamb year was slow-growth or tended to be stable over time. For Saint Croix hair sheep, lambing years had a significant (*p* < 0.05) effect on litter size and weight at weaning [46]. The genetic correlation between birth weight and weaning weight was 0.44, moderately positive, and phenotypic correlations were similar to the genetic correlation [47]. The higher the correlation, the greater the degree of improvement. The strong positive genetic correlation between traits indicated that the selection of favorable or unfavorable traits would lead to the genetic change of related traits. Estimation of heritability provides information about the extent to which a particular genetic character is transmitted to offspring. These genetic and phenotypic correlations, along with heritability, supply the basis for the more accurate prediction of the achievement of intricate sheep breeding programs aimed at improving several traits.

### 4.2. Genome-Wide Association Study

Reproductive traits are complex traits that are regulated by multiple factors. The heritability of LS in Xinggao ewes, assessed by genetic parameters, was higher than in previous studies. Genome-wide association studies help to find genetic variants linked to one or more interesting phenotypes [48]. Thus, it is necessary to explore SNPs and genes associated with LS. GWAS is a useful method to identify linked loci and candidate genes through the analysis of the association between the genotypes and the phenotypes of individuals [49]. Novel SNPs with modest positive signals can be discovered in GWAS when sample sizes are insufficient [50]. Although we had fewer samples, which may affect the number of prominent SNPs, the Q-Q plots showed the inflation factor to be 1, implying that the chance of a false positive was small. Georgiopoulos et al. [51] reported the size of the inflation factor affects the statistical ability, and the relatively smaller inflation factor does not require significant adjustment of the sample size. Others have been reported 47 Pelibuey sheep with 3 SNPs showing strong significance with litter size, and 54 SNPs were only suggestive on GWAS analysis [52]. The two suggestive significant SNPs (oar3_OAR14_38652542, OAR25_39236034.1) were associated with LS (Table 3), and the Q-Q plot (Figure 3A(b)) showed the inflation factor to be 1.002, indicating that the chance of a false positive was small. The *PHLPP*2 gene is a type of serine/threonine phosphatase that has been identified in cells and is involved in the regulation of AGC kinase and plays an important role in cell signal transduction and cell function regulation [53]. The *TAT* (tyrosine amino transferase) gene is assigned to chromosome 14 and spans 9995 bp composed of 12 exons and is expressed in the liver, kidney, brain, oviduct, and ovary. *TAT* is an estrogen-dependent gene that can be used in hormone therapy in the monitoring and treatment of ovarian carcinomas [54,55] and significantly differentially expressed at different stages of oviposition in black Muscovy ducks, in which it may be involved in the regulation of ovarian development and may help regulate ovarian maturation and egg production [56]. 

For growth traits, we found two SNPs (oar3_OAR15_41596318, Chr6:114773309), three SNPs (OAR14_22829882.1, OAR21_12710366.1, oar3_OAR12_49504727), and three SNPs (oar3_OAR14_58915171, OAR14_22829882.1, oar3_OAR12_49504727) associated with BWT, WWT, and ADG, respectively, and mapped near or within regions of genes (*MRVI*1, *LYVE*1, *RNF*141, *DOK*7, *RGS*12;*DLG*2, *AGRN*, ISG15, *PERM*1, *PLEKHN*1, *KLHL*17, *NOC*2*L,SAMD*11, and *NLRP*9) (Table 3) that were known to play an important role in growth and development. Among these genes, the *DOK*7 gene was underlined as a member of the kinase (DOK) protein family, which involves in intracellular signal transduction pathways downstream of receptor tyrosine kinases (RTKs) [57]. *DOK*7 inhibits the proliferation, migration, and invasion of breast cancer cells through the PI3K/PTEN/AKT pathway [58]. *RGS*12 is a regulator of G protein signaling (RGS proteins), and GTPase-accelerating proteins (GAPs) in cells attenuate G protein-dependent signals that are received by cells from the external environment [59]. *RGS*12 expression is temporally and spatially regulated in developing mouse embryos and is significantly expressed in somites and developing skeletal muscles and also expressed in primary myoblasts [60]. *DLG*2 (Disks Large Homolog 2) as a tumor suppressor candidate, of which its structural mutation is associated with osteosarcoma, and *DLG*2 deletion accelerates the growth of bone tumors in canine and human cell lines [61]. The mammalian DLG protein family is mainly known for its role in epithelial polarity and polarity during cell division [62], and *DLG*2 was pointed towards the hippo signaling pathway; this pathway is evolutionarily conserved and it controls organ size by regulating cell proliferation, apoptosis, and stem cell self-renewal [63]. *NLRP*9 (NLR family pyrin domain containing 9) is one of the *NLRP* gene family genes with replication and functional diversity in the mammalian reproductive system, especially in reproductive functions such as the specific amplification of *NLRP*9 in mice [64]. Of the 14 human *NLRP* genes, 10 NLRPs (2, 4, 5, 7, 8, 9, 11, 12, 13, 14) were detected in oocytes and/or early embryos [65]. Moreover, localization of NLRP9B protein is detected in mouse oocytes, and, interestingly, the NLRP9B protein decreased and correlated with oocyte aging [66].

We noted that previous studies have identified several genes that mainly affect high fecundity in ewes, such as *FecB*, *BMP*15, and *GDF*9. Compared with previous works, we detected a novel set of genes related to reproduction and growth near the SNP. The main reason may be that most of the earlier studies were based on genome-wide selection attempts on prolific and non-prolific breeds with low SNP densities, possibly due to the weak ability to detect this association when dealing with traits of interest quantitatively in a small sample size. In addition, these sheep may be selected by environmental variables, such as climate, diet, and so on. Generally, in Xinggao sheep, all these genes may play an important role in regulating breeding.

## 5. Conclusions

In summary, this study aimed to compile the heritability and GWAS related to re(productive) traits in Xinggao sheep and attempted to highlight possible disciplines that are expected to benefit local sheep breeding. This was the first study on genetic parameters in Xinggao sheep for reproduction and growth. Estimates of heritability were low to moderate. Genetic correlations between traits were positive and generally high, and genetic and phenotypic correlations between these traits are favorable. The estimated genetics parameters indicate that the LS and weight in Xinggao sheep could be genetically improved. Therefore, weight traits or related traits can be used to select growth-promoting breeds. Additionally, our results suggest a differential genetic regulation mechanism for a novel set of genes involved in local sheep reproductive and growing functions. These novel MTAs have important applications in future molecular breeding and in providing important insights into the regulation of breeding in sheep and other mammals.

## Figures and Tables

**Figure 1 genes-14-00579-f001:**
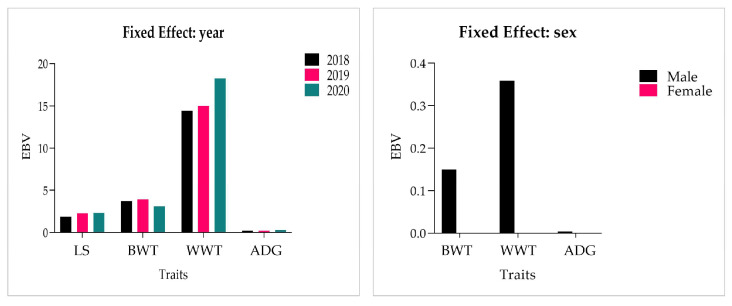
EBV for different traits in fixed effects.

**Figure 2 genes-14-00579-f002:**
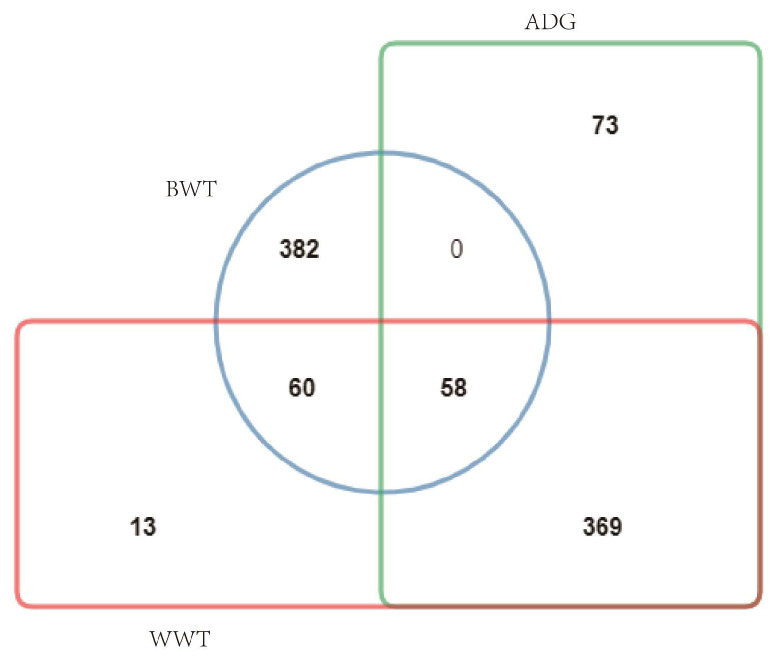
Venn network about 3 traits of lambs in top 500 EBV.

**Figure 3 genes-14-00579-f003:**
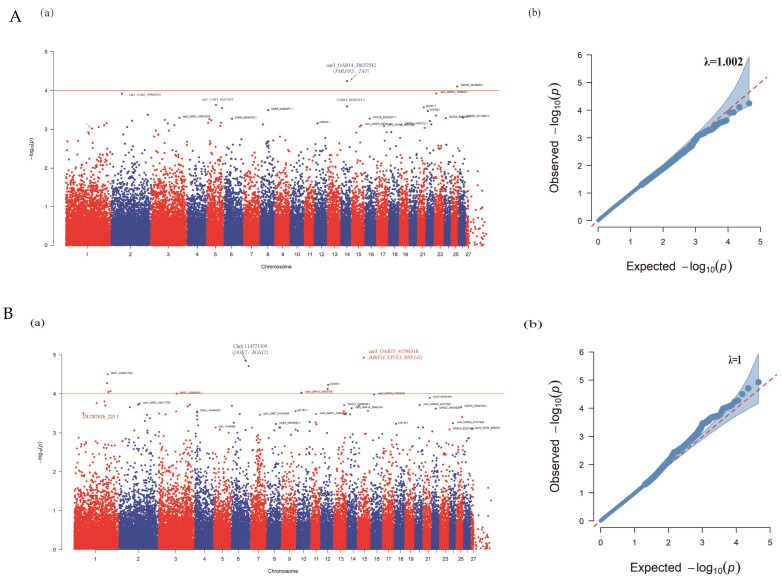
Manhattan and quantile–quantile (Q-Q) plot to SNPs associated with traits on the ovine chromosome. (**A**) LS; (**B**) BWT; (**C**) WWT; (**D**) ADG. (**a**) Manhattan plots X-axis are SNPs positions on chromosomes, Y-axis is -Log10 *p*. The red line corresponds to the threshold for *p* < 10^−4^ statistically suggestive significance. (**b**) Q-Q plots of the traits drawn by the expected P and observed *p* of each SNP.

**Table 1 genes-14-00579-t001:** Descriptive statistics for ewe reproduction and lamb performance.

Item	Ewe Reproduction	Lamb Performance		
	LS	BWT, kg	WWT, kg	ADG, kg
Mean	2.12	3.51	16.19	0.25
S.D.	0.64	0.77	3.49	0.07
Minimum	1	1.5	6.67	0.05
Maximum	5	7	25.93	0.47
Number of records	2289	2281	2284	2284

LS (litter size), BWT (birth weight), WWT (weaning weight), ADG (average daily gain), S.D. (standard deviation).

**Table 2 genes-14-00579-t002:** Heritability (bold on diagonal), genetic correlation (above diagonal), and phenotypic correlation (below diagonal).

Trait	Ewe Reproduction	Lamb Performance		
	LS	BWT	WWT	ADG
LS	**0.12 ± 0.00**	-	-	-
BWT	-	**0.22 ± 0.02**	0.34 ± 0.44	0.12 ± 0.48
WWT	-	0.38	**0.28 ± 0.05**	0.97 ± 0.00
ADG	-	0.13	0.97	**0.29 ± 0.06**

LS (litter size), BWT (birth weight), WWT (weaning weight), ADG (average daily gain), SE (standard error).

**Table 3 genes-14-00579-t003:** Significantly associated SNPs with the re(production) traits in Xinggao sheep.

Traits	SNP ID	Chr	Position(bp)	Gene Annotated	*p*
LS	oar3_OAR14_38652542	14	38,652,542	*PHLPP*2, *TAT*	5.70 × 10^−5^
	OAR25_39236034.1	25	39,236,034	-	7.77 × 10^−5^
BWT	**oar3_OAR15_41596318**	**15**	**41,596,318**	***MRVI*1, *LYVE*1, *RNF*141**	**1.18 × 10^−5^**
	**Chr6:114773309**	**6**	**114,773,309**	***DOK*7,** ***RGS*12**	**1.95 × 10^−5^**
WWT	**OAR14_22829882.1**	**14**	**22,829,882**	**-**	**2.37 × 10^−5^**
	**OAR21_12710366.1**	**21**	**11,143,699**	***DLG*2**	**2.49 × 10^−5^**
	**oar3_OAR12_49504727**	**12**	**49,504,727**	***AGRN*,*ISG*15, *PERM*1, *PLEKHN*1, *KLHL*17, *NOC*2*L*, *SAMD*11**	**2.72 × 10^−5^**
ADG	**oar3_OAR14_58915171**	**14**	**58,915,171**	***NLRP*9**	**4.56 × 10^−5^**
	**OAR14_22829882.1**	**14**	**22,829,882**	**-**	**1.40 × 10^−5^**
	**oar3_OAR12_49504727**	**12**	**49,504,727**	***AGRN*,*ISG*15, *PERM*1, *PLEKHN*1, *KLHL*17, *NOC*2*L*, *SAMD*11**	**2.59 × 10^−5^**

Genome-wide significant associations are in bold.

## Data Availability

The datasets of genotypes analyzed during the current study are available on figshare (DOI: 10.6084/m9.figshare.22153964). The phenotypic data is not publicly available as the populations consist of the nucleus herd of Dumei Animal Husbandry Biotechnology, but are available from the corresponding author on reasonable request.

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
