# Peer review of "Estimates of Genomic Heritability and the Marker-Derived Gene for Re(Production) Traits in Xinggao Sheep"

_genes, 2023, doi:10.3390/genes14030579_

Round 1

Reviewer 1 Report

Manuscript Number: GENES-2213408

Title: Estimates of genomic heritability and the marker-derived gene for re(production) traits in Xinggao sheep

The work is a good attempt at trying to estimate the genetic parameters for four re(production) traits in Xinggao sheep. The authors performed a genome-wide association study (GWAS) to identify novel marker-trait associations (MTAs) with prolificacy and growth. 

The results are not of outstanding scientific importance. The work is ACCEPTABLE, but must be improved. Some comments and suggestions are addressed to the authors.

Main comments

Much of the discussion only compared findings with other papers for the studied traits and sometimes repeated data from the results was observed, especially in topic 4.1.  Regarding to topic 4.2, it was not discussed why it was not found any SNPs near the several FEC genes already described in the literature associated with prolificacy in sheep. The discussion should be improved. 

Extensive editing in English language and style is needed

Standardize the references throughout the paper.

In the conclusions the authors did not mention anything about the genetic parameter estimates.

Specific Comments

1- The sentence  is confusing (lines 37-40) 

2- There is a misconception between fertility and prolificacy. Line 70, the word fertility should be replaced by prolificacy. 

3- Standardize some terms, e.g., "litter size" sometimes appears as "litter number" (line 66).

4- Line 85, replace "high rate of reproduction rate" by "high reproduction rate".

5- Improve the sentence "Phenotypes were obtained records of Xinggao ewes and lambs" (line 102).

6 - Replace the sentence "the mean number of LS" by "the mean of LS" (line 107).

7- There is no need to present the histogram. Information is clear in the text. 

8 - Remove the sentence "This is a table/figure" (lines 110, 124, 225, 249, 251, 268 and 273).

9 - Replace the sentence "pupping season" by "lambing season" (line 134).

10- The sentence  (lines 165-168) is repeated (line 149-152). Lines 165-168, the meaning of the formula of h2 to BWT, WWT and ADG should be presented.

11 - Line 175, It should mention the characteristics to which the collection of blood from 46 lambs refers. 

12 - Line 176, it is microliter, not milliliter.

13 - It should be 95 samples (49+46), not 96.

14 - For better understanding, list using numbers the filtering criteria used (lines 185, 187, and 188).

15- Improve the sentence (lines 188-191), it is confusing. 

16- Line 209, remove the sentence "50kb SNPs".

17- Line 213, the information does not match to the figure. In the figure 936 ewes had 2 or more offspring. 

18- Line 214-215, most lamb weigh at birth were between 2.5 and 4.0kg, and the weaning weight were between 12 and 15kg.

19- Improve and summarize figure 3. Leave only the first part.

19 - Line 288, what does "Line S Fertility" mean?

21 - Line 323, improve the expression "observed by me".

Author Response

Question1: Much of the discussion only compared findings with other papers for the studied traits and sometimes repeated data from the results was observed, especially in topic 4.1. Regarding to topic 4.2, it was not discussed why it was not found any SNPs near the several FEC genes already described in the literature associated with prolificacy in sheep. The discussion should be mproved.

Response1: I revised discussion of 4.1 and 4.2, and in line 396 to 401 disscued why it was found SNP near Fec genes: We noted that previous studies have identified several genes that mainly affect high fecundity in ewes, such as FecB, BMP15, and GDF9. Compared with previous, we detected a novel set of genes related to reproduction and growth near the SNP. The main reason may be that most of the earlier studies were based on genome-wide selection attempts on prolific and non-prolific breeds with low SNP densities, possibly due to the weak ability to detect this association when dealing with traits of interest as a quantitative in a small sample size. In addition, these sheep may be selected by environmental variables, such as climate and diet, and so on. Generally, in Xinggao sheep, all these genes may play an important role in regulating breeding.

Question4: In the conclusions the authors did not mention anything about the genetic parameter estimates.

Response4: We have added the last part of the abstract, please check it at L408~L413. This was first study on genetic parameters in Xinggao sheep for reproduction and growth. Estimates of heritability were low to moderate. Genetic correlations between traits were positive and generally high, genetic and phenotypic correlations between these traits are favorable. The estimated genetics parameters indicate that the LS and weight in Xinggao sheep could be genetically improved. Therefore, weight traits or related traits can be used to select growth-promoting breeds.

Question5: The sentence is confusing (lines 37-40)

Response5: Thank you for your suggestions. I had revised that China’s local breeds have the advantages of crude feed tolerance, strong adaptability, and low maintenance requirements, but the growth rate is slow, the reproductive performance is poor, the performance measurement is lagging, the breeding intensity is not enough, and the population genetic progress is slow.

 Question 6:There is a misconception between fertility and prolificacy. Line 70, the word fertility should be replaced by prolificacy.

Response6: We are very sorry for our misconception. As reviewer suggested that fertility is modified to prolificacy in manuscript.

 Question 7: Standardize some terms, e.g., "litter size" sometimes appears as "litter number" (line 66).

Response 7: Thank you for your suggestions. I have revised the full text for standardizedd terms in the final manuscript.

 Question 8: Line 85, replace "high rate of reproduction rate" by "high reproduction rate".

Response 8: Thank you for your suggestions. I had revised it.

Question 9: Improve the sentence "Phenotypes were obtained records of Xinggao ewes and lambs" (line 102).

Response 9: Thank you for your suggestions. It was modified to: Phenotypes records for Xinggao ewes and lambs were gathered

Question 10: Replace the sentence "the mean number of LS" by "the mean of LS" (line 107).

Response 10: Thank you for your suggestions. I revised it in the manuscript.

Question 11: There is no need to present the histogram. Information is clear in the text. 

Response 11: Thank you for your suggestions. I have deleted Figure 1.

Question 12: Remove the sentence "This is a table/figure" (lines 110, 124, 225, 249, 251, 268 and 273).

Response 12: Thank you for your suggestions. I have removed in Line 110, 225, 249, 251, 268, and 273.

Question 12: Replace the sentence "pupping season" by "lambing season" (line 134).

Response 12: Thank you for your suggestions. I revised it in the manuscript.

 Question 13: The sentence (lines 165-168) is repeated (line 149-152). Lines 165-168, the meaning of the formula of h2 to BWT, WWT and ADG should be presented.

Response 13: Thank you very much for your careful writing advice, I have changed to where h2 is heritability; σ2a is direct additive genetic variance; σ2pe is permanent environment variance related to repeats; σ2m is maternal genetic variance, and σ2e is residual variance.

Question 14: Line 175, It should mention the characteristics to which the collection of blood from 46 lambs refers.

Response 14: Added: Among 46 healthy and active lambs, blood samples collected showed a selective birth weight ranged from 2.3 kg to 6.3 kg, weaning weight ranged from 10.2 kg to 21.51 kg, and average daily gain ranged from 0.14 kg to 0.37 kg

Question 15: Line 176, it is microliter, not milliliter.

Response 15: I am sorry for my careless writing, I have changed it to ul.

Question 16: It should be 95 samples (49+46), not 96.

Response 16: I am sorry for my careless writing, I have changed it to 95.

Question 16: For better understanding, list using numbers the filtering criteria used (lines 185, 187, and 188).

Response 16: Thank you for your suggestions. 51,867 SNPs and individuals with call rate < 95% (-geno:0.05 and -mind:0.05) and mi-nor allelic frequency (MAF) < 0.01 and failed the Hardy-Weinberg Equilibrium (HWE) (P < 1e-6) were excluded [29]. Finally, 45,147 SNPs from 49 ewes were used to carry out genotype association tests, and for lamb growth traits: BWT, WWT, and ADG were performed on 45,147 SNPs distributed in 27 chromosomes, and these SNPs that were positioned on the chromosome (1 to 27).

Question 17: Improve the sentence (lines 188-191), it is confusing.

Response 17: I have modified it to: Finally, genotype association tests were performed on 45,147 SNPs distributed on 27 chromo-somes in 49 ewes and 46 lambs.

Question 18: Line 209, remove the sentence "50kb SNPs".

Response18: I am sorry for my careless writing, I have deleted it.

Question 19: Line 213, the information does not match to the figure. In the figure 936 ewes had 2 or more offspring.

Response 19: There is a mistake in writing here. According to the statistics in Table 1, 946 ewes gave birth to an average of 2.12 lambs; only 151 ewes gave birth to a single lamb, and the rest gave birth to two or more lambs (Figure 1)

Question 20: Line 214-215, most lamb weigh at birth were between 2.5 and 4.0kg, and the weaning weight were between 12 and 15kg.

Response 20: Thank you for your suggestions. I have revised it in the manuscript.

 Question 21: Improve and summarize figure 3. Leave only the first part.

Response 21: I have removed the excess, leaving the first part.

Question 22: Line 288, what does "Line S Fertility" mean?

Response 22: Line S Fertility mean that beginning and end of the joining period for selection (S) lines in each phase of the Masood’s study.

 Question 23: Line 323, improve the expression "observed by me".

Response 23: Thank you for your suggestions. I revised to: Analysis of estimated breeding values (EBVs) for BWT, WWT, and ADGshowed that the sex had differential effects, with males being  heavier than females for all three traits. The direct autosomal heritabilities for birth weight and weaning weight was 0.12 and 0.29,respectively; and the sex effect was significant (P < 0.01) across all traits [44], similar result was observed.

Reviewer 2 Report

The authors Estimated genomic heritability and the marker-derived gene  for re(production) traits in Xinggao sheep

I suggest extensive English editing of the manuscript. All sections are poorly described

Collection of blood samples was carried out. I did not find ethical approval

Materials and results sections should be improved in their presentation

Figure 1 is of low quality. Therefore, it should be improved. The authors cited that Figures 1, 2 and 3. This is a figure. Ewe reproduction and lamb performance traits frequency distribution 124 histogram. What is meant by This is a figure?

The identified regions by genome wide analysis demonstrated genes associated with the studied traits. I suggest providing more information about these genes in the discussion section.

Author Response

Question: I suggest extensive English editing of the manuscript. All sections are poorly described.

Collection of blood samples was carried out. I did not find ethical approval

Materials and results sections should be improved in their presentation

Figure 1 is of low quality. Therefore, it should be improved. The authors cited that Figures 1, 2 and 3. This is a figure. Ewe reproduction and lamb performance traits frequency distribution 124 histogram. What is meant by This is a figure?

The identified regions by genome wide analysis demonstrated genes associated with the studied traits. I suggest providing more information about these genes in the discussion section.

Response:

In response to your questions, we would like to give you the following answers:

I have revised the English language extensively in the manuscript.

I have added the ethical approval in manuscript: The animal study was reviewed and approved by the Special Committee on Scientific Research and Academic Ethics of Inner Mongolia Agricultural University [Approval No. (2022)0903].

Combined with the suggestion of reviewer 1 and the result of my article, the frequency of traits has been clearly described in the article, so Figure 1 is deleted.

I've added a discussion of genes in L388~395.

Round 2

Reviewer 2 Report

The authors have responded to my comments. I suggest acceptance of the manuscript in its current form